# Quantifying Decay Due to Wet Atmospheric Deposition on Basalt

**DOI:** 10.3390/ma16165644

**Published:** 2023-08-16

**Authors:** Luis Miguel Urbina Leonor, Rodolfo Sosa Echeverría, Ana Luisa Alarcón Jiménez, Mónica Solano Murillo, Graciela Velasco Herrera, Nora A. Perez

**Affiliations:** 1Posgrado de Ingeniería Ambiental, Universidad Nacional Autónoma de México, Mexico City 04510, Mexico; luis.miguel.urbina.leonor@gmail.com; 2Instituto de Ciencias de la Atmósfera y Cambio Climático, Universidad Nacional Autónoma de México (ICAyCC-UNAM), Mexico City 04510, Mexico; ana.alarcon@atmosfera.unam.mx (A.L.A.J.); monica@atmosfera.unam.mx (M.S.M.); 3Instituto de Ciencias Aplicadas y Tecnología, Universidad Nacional Autónoma de México, Mexico City 04510, Mexico; graciela.velasco@icat.unam.mx; 4CONACyT—Laboratorio Nacional de Ciencias para la Investigación y Conservación del Patrimonio Cultural, Instituto de Investigaciones Estéticas, Universidad Nacional Autónoma de México, Mexico City 04510, Mexico; norari.perez@gmail.com

**Keywords:** volcanic stone, stone heritage, stone decay, heritage weathering, built heritage, wet atmospheric deposition

## Abstract

The study of building materials is important for a better conservation of built heritage. Worldwide, volcanic stones (including basalt, andesite and dacite) are among the least studied building materials. In this research, the decay of a red basalt due to wet atmospheric deposition was studied. Red basalt was exposed to artificial rain solutions, prepared from rain samples collected weekly from 2014–2019. In this research, the decay of stone-built heritage was indirectly studied emulating wet atmospheric accelerated weathering under three different volume weighted mean (VWM) compositions: global, acid and no-acid categories. Lixiviates were analyzed to better understand the deterioration mechanisms taking place inside the material. Decay was quantified as mass difference, water absorption capacity (WAC) and open porosity (OP) changes. Results show that the methodology used is suitable to research the decay of built heritage. The studied basalt is indeed prone to decay by wet atmospheric deposition. The main decay mechanisms are the washing of insoluble compounds, dissolution of minerals, salt crystallization and cation exchange. WAC and OP showed promising results of their appropriateness as monitoring variables of decay in situ. Acid conditions produce the most severe decay, but Ph effect is not as important as precipitation volume. Non-linear equations relating volume of precipitation with mass difference in red basalt are presented.

## 1. Introduction

The influence of atmospheric agents on the decay of heritage buildings is a topic that has been studied for over 60 years [1]. Throughout the decades, knowledge on the subject has improved to a great extent (i.e., [2,3,4]). This knowledge, although still insufficient to guarantee any adequate conservation efforts, has produced important advances, such as the study of different materials, as in [5,6,7,8,9,10], the engineering of damage functions (also known as dose-response, i.e., [8,11,12,13,14]), the development of coating materials (for example, [15,16,17,18]) or the modelling of microclimatic conditions (as [19,20,21]). Early research focused on wet atmospheric deposition, while in later study, more complex mechanisms such as dry deposition [22], salt crystallization [23] or thermal fatigue [24] have all been explored. “Wet atmospheric deposition” refers to the settling of different compounds present in the atmosphere, both gases and particles, through precipitation (as rain, fog or snow) [25].

Natural stones are one of the main building materials for construction. They are used following different criteria: availability, aesthetic value and even cost [26]. Most of the published work on the matter has studied European building materials, especially carbonate stones [8,18,27,28,29,30], and occasionally, mortar or granite [31,32,33,34]. However, volcanic stones have been and are still used as building materials worldwide: in Europe [35,36,37], Africa [38], America [5,39,40,41,42], Asia [43] and Oceania [44].

To ensure preservation of built property worldwide, building materials need to be scientifically studied. This research was done specifically to increase knowledge on volcanic stone-based heritage buildings. Here, the effect of wet atmospheric deposition on building decay is discussed. This first approach to decay understanding can be complemented in the future, with multidimensional research also involving meteorology, material nature and air quality. In contrast to universally known materials (such as the Carrara marble), volcanic stones have not been studied rigorously, even though they were often used in important cultural sites worldwide, such as in Perú, South Pacific or Ethiopia. However, several volcanic-stone-based ensembles are in areas where few atmospheric ground-based observations (for this article, “atmospheric” refers to the combination of meteorology, wet atmospheric deposition and air quality conditions) are available. In those cases, damage functions can be useful tools. Damage functions are equations that relate certain atmospheric conditions (such as precipitation volume, pollutants concentration and air temperature) with a certain quantifying decay variable, usually mass loss (commonly presented as “surface recession” for metamorphic carbonate stones [8]) for a specific material. Development of damage functions is important because they model decay, but also because they can be useful to have a rough idea of future decay of the building material. Even if there is no atmospheric ground-based data, a general idea of the areas in which building assets are prone to decay can be computed, and hence, decision makers can have a clearer perspective on where to focus resources on the endeavor for conservation [45].

In Mexico, volcanic stones constitute an important family of building materials. They were used in several UNESCO heritage sites as the main building materials, such as the Historic Centers of Mexico City, Morelia, Oaxaca or San Luis Potosí [5,39,41]. Among the volcanic stones, the main minerals include basalt, ignimbrites, andesites, trachyandesites and trachydacites. One of the most important materials for its widespread use as architectural finishes is a red basalt, known as “tezontle”, described as a “well-cemented red agglomeratic scoriaceous rock” [46]. Tezontle is an important building material in all of Mexico City’s Historical Center [39,41], and was employed in numerous buildings, both from the Pre-Hispanic and Colonial periods including the Templo Mayor, the Metropolitan Cathedral, The Palace of the Counts of Santiago de Calimaya, and Saint Michael the Archangel church, to mention a few examples (Figure 1).

In this article, the first few steps towards a damage function for a volcanic stone are made, focusing on the effect of wet atmospheric deposition in a basalt. Thus, the general scope of this work is to study basalt deterioration through micro-destructive analysis. The main decay mechanisms are observed, the variables controlling them are identified, the effect of wet atmospheric deposition on red basalt is quantified, the effect of different chemical compositions of rain are compared and two different in situ monitoring variable alternatives to mass loss are examined: water absorption capacity (WAC) and open porosity (OP).

## 2. Materials and Methods

Four different aspects were studied. The first was the analysis of wet atmospheric deposition collected from 2014–2019 in a station in the Historical Center of Mexico City; the second was the measuring of water absorption properties (WAC, OP) of the building material before and after the accelerated weathering test. Additionally, accelerated weathering was tested, and lixiviated samples from the accelerated weathering experiment were analyzed by ion chromatography (IC). Finally, all the data underwent a statistical treatment. The whole experimentation is summarized in Figure 2.

The studied material is a red basalt (“tezontle”). The expected composition of the rock has been reported as mainly quartz and plagioclases, with some ferromagnesian compounds [47]. The stone was obtained from the same supplier used by the Restoration Department at the Museo del Templo Mayor (Templo Mayor Museum), “Santa Maria Tepepan” materials, Mexico City, Mexico, when an intervention is needed. Specimens were obtained from a single stone ashlar with a diamond blade hand grinder without lubricant to avoid contamination and washing of clays and salts. Basalt possesses a highly heterogeneous porosity; consequently, cutting specimens with the same mass was very difficult. Instead, all the specimens have the same dimensions (approximately 5 cm × 5 cm × 2 cm; Figure 3) to have an equal apparent volume. It is important to clarify, for the purposes of this article, that “specimen” will refer to natural stone pieces, whereas “samples” will refer to liquid collected materials.

### 2.1. Analysis of Wet Atmospheric Deposition

The Sección de Contaminación Ambiental of the Instituto de Ciencias de la Atmósfera y Cambio Climático of the Universidad Nacional Autónoma de México (Environmental Pollution Group of the Atmospheric Sciences and Climate Change Institute of the Mexico’s National Autonomous University, SCA-ICAyCC-UNAM) analyzed the wet atmospheric deposition samples in collaboration with the Sistema de Monitoreo Atmosférico of the Gobierno de la Ciudad de México (Atmospheric Monitoring System of Mexico City’s Government, SIMAT) during the rainy season (approximately from May to October). The procedure is based on the one used by the World Meteorological Organization (WMO): using a Model TE-78-100 Tisch precipitation collector (Tisch Environmental, Cleves, OH, USA) that samples only wet atmospheric deposition during a whole week. SIMAT personnel measured the collected rain volume. Samples greater than 60 mL (2 mm of rain), were collected on previously washed bottles, with electric conductivity (EC) lower than 1 μS cm^−^¹. Samples were then filtered with a 0.2 micrometer membrane (Figure 3b) in the SCA-ICAyCC-UNAM laboratory and stored at 4 °C until analysis. The SIMAT has a network of 16 stations, and since basalt, both black and red, is an important building material in the Historical Center of Mexico City, the station Museo de la Ciudad de México (Mexico City’s Museum, MCM), located in the Palace of the Counts of Santiago de Calimaya, was selected as representative for the area because it is part of the UNESCO declaratory document [48].

### 2.2. Water Absorption Properties

When studying rocks used as building materials, different micro-destructive analytical techniques are common (i.e., Optical Microscopy of thin section, X-Ray Diffraction, X-Ray Fluorescence, Carbon Fraction [2]). However, since direct measuring of decay on built property is not really feasible, the decay can only be monitored through estimations of mass loss. In this study, two properties are proposed as decay monitoring variables: WAC and/or OP, which can be easily measured with a Karsten tube [48]. These analyses were performed following the method of ICCROM [49]. This test was done only before and after the accelerated weathering test, to avoid depletion of possible recrystallized salts, the washing of possible decay products and insoluble compounds (especially clays) and in general, additional decay.

### 2.3. Accelerated Weathering

An accelerated weathering test was performed in the chamber of wet atmospheric deposition accelerated tests built by [9] (Figure 4). To irrigate the sample, an artificial rain solution was irrigated above the tezontle specimens, and both the leaching solutions and the irrigated artificial rain irrigated were collected.

Previous studies have shown that the minimum simulation time for a perceptible decay on building materials is 10 years [9,34], so the specimens were exposed to the equivalent volume of artificial rain. The important role of rain pH on stone decay has been widely discussed by several authors (i.e., [8,9,11,28]). “Clean” rain has a pH of 5.6, while “acid rain” has a lower pH [50]. Hence, the rain samples collected in the “Analysis of wet atmospheric deposition sampled” were classified in three different groups. The first group considers the totality of the samples (“global”); the second one only includes rain samples whose pH was lower than 5.6 (“acid”); and the third considers samples with a pH of 5.6 or higher (“no acid”). The volume of global precipitation is the aggregate of the acid and no acid precipitation.

A major challenge the study of cultural heritage faces is that accessible samples are limited. However, when working with the same building material that does not belong to a cultural property, it is possible to treat the results statistically. To compare the effect of each class of rain, an ANOVA comparison was performed between the three treatments with at least 6 specimens with a significance level (α) of 0.1 and an effect size of 0.9 [51].

To calculate the equivalence from volume of precipitation to years, the results from rain samples for MCM station in the period 2014–2019 were used. An annual average was calculated for each class of rain. The amount of rain that would precipitate in the exposed area of the specimens (approximately 25 cm^2^) per year was calculated using Equation (1).
(1)$$0.0025\;m^{2}\cdot R\frac{L}{m^{2}\cdot year}=0.0025R\;\frac{L}{year}$$
where *R* stands for the annual mean precipitation (in mm) for each rain type.

Before the beginning of the test, the volume irrigated by each dripping line was set to 30 mL/min, with the help of a graduated cylinder and a chronometer for all precipitation groups.

One cycle in the accelerated weathering chamber was the irrigation of a certain volume per specimen (4.88 L for global precipitation; 3.82 L for the no-acid group and 1.06 L for acid class) with a temperature of 16 °C, relative humidity 50% and no radiation applied; afterwards, the specimens were left to natural drying (that is, to laboratory conditions, between 16–23 °C and 40–65% RH) for 20 h. Each cycle simulates the drying-wetting conditions that the specimen would be exposed to for two years. This process was repeated until all the cycles were finished. To simulate 10 years of exposure, 5 cycles were performed on the weathering chamber. The number of cycles was selected to have six observations and allow salt crystallization inside the material. Total volume corresponds to the total amount of artificial rain prepared for all 6 samples and is presented in the “Analysis of wet atmospheric deposition” section. Hence, the irrigation lasted for about 164 min for global precipitation, 129 min for no acid and around 37 min for acid group. In all cases, immediately after the volume of rain finished irrigation, specimens were dried with a microfiber cloth and weighed, to obtain the partial-wet weight. On each of these experimentation cycles, the mass of the specimens was measured, to monitor decay. This measurement is not conclusive, because even if the specimens were apparently wet, it is not possible to guarantee that they reached water saturation. In the results, section “volume of artificial rain” refers to the total volume of solution irrigated on each specimen. This is important because although 10 years are simulated in all cases, global precipitation volume is always higher than acid and no-acid cases.

Three artificial solutions of rain were prepared, one per category: global, acid and no acid. Previous studies have shown that in addition to pH, the chemical composition of rain is important for its effect on any material [52]. However, in previous accelerated weathering publications, the rain has been prepared either with worst-case scenarios [9] or using the arithmetic average of samples collected monthly [52]; hence, they both produce non-representative mass loss results. The worst-case scenario will overestimate the decay because conditions are regularly not that severe. On the other hand, the arithmetic average assumed that all events produced the same impacts regardless of their volume. For example, if 5 mm of rain were collected in a month, and 30 mm were sampled in another month, both chemical compositions will be considered equally. To avoid this, the volume weighed mean (VWM) was used: this consideration will give more importance to those samples whose rain volume is greater and diminishes the importance of rain samples with lower volume.

The chemical composition using VWM in μEq L^−^¹ was calculated for Cl^−^, NO₃^−^, SO₄^2−^, K⁺, Na⁺, NH₄⁺, Mg^2^⁺, Ca^2^⁺ and H⁺ ions with Equation (2) (exemplified for H⁺).
(2)$${\bar{X}}_{\left[H^{+}\right]}=\frac{\sum_{i=1}^{n}{\left[H^{+}\right]}_{i}\cdot {V_{rain}}_{i}}{\sum_{i=1}^{n}{V_{rain}}_{i}}$$
where $\bar{X}$ is an ion in the sample (μEq L^−^¹); ${\left[H^{+}\right]}_{i}$ represents the concentration of H⁺ in the *i* sample (μEq L^−^¹); ${V_{rain}}_{i}$ the volume measured for sample *i* (mm).

After the VWM were obtained for each rain class and knowing the volume of artificial rain needed per specimen (Equation (1)), the total amount of solution needed to simulate 10 years on 6 specimens was calculated (Table 1), as well as the compounds needed to recreate each kind of rain. A Spearman rank correlation was calculated to know the relationship between ions. Then, similar ions were paired to know the needed compounds to recreate the objective composition. If there was still a lack of the ion, it was paired with another one. The resulting amount of each compound needed, depending on if they were solid or liquid, classified according to artificial rain pH, are presented in Table 1.

To minimize chemical composition variation, 2 L of solution with all the compounds needed was prepared for each kind of deposition. Then, 500 mL of that solution were mixed in the volume needed to simulate 2 years of precipitation, and finally, the pH was corrected either with H_2_SO_4_ 0.01 M or NH_4_OH 0.01 M, which were the more abundant ions.

Samples of the solution that interacted with the stone specimens (“lixiviate samples”) and the irrigated solution were collected at the beginning of the first accelerated weathering test, and afterwards when there was 500 mL left for the volume to finish each cycle, in bottles whose EC was lower than 1 μS cm^−^¹. Additionally, a sample of the artificial rain solution that did not interact with the specimen was collected (“blank”) for a total of 42 lixiviate samples per precipitation group. The initial sample was collected to test for possible remaining salts after the water absorption properties test.

### 2.4. Ion Chromatography (IC)

A total of 152 rain samples were collected and analyzed using the IC, as well 126 aqueous samples from accelerated weathering were analyzed (42 per rain class) including 6 “blanks” (the solution of artificial rain irrigated without interaction with stone specimens) were collected one per accelerated weathering cycle.

For the chemical analysis, samples were vacuum-filtered with a 0.22 μm membrane. Then, EC was measured with a YSI 32 Yellow Spring Instrument, Co. Inc. (Yellow Spring, OH, USA) and HORIBA D-424 (Horiba Ltd. Company, Kyoto, Japan) conductivity instruments. Alkalinity was calculated with the Gran titration method [53] using a Metrohm 827 (Metrohm AG, Herisau, Switzerland) and Orion 960 (Vernon Hill, IL, USA) pH meters. The main soluble ions were measured by IC. Anions (Cl^−^, NO_3_^−^, SO_4_^2−^) were measured in a Metrohm 883 IC (Metrohm AG, Herisau, Switzerland) with chemical suppression (as US-EPA Method 300.1 stipulates), using a Metrosep A Supp 4–250/4.0 column (Metrohm AG, Herisau, Switzerland). Cations (K⁺, Na⁺, NH_4_⁺, Mg^2^⁺, Ca^2^⁺) were analyzed with an isocratic Waters 510 pump (Waters corporation, Milford, MA, USA), a Waters 432 conductivity detector (Waters corporation, Milford, MA, USA) and a Metrosep C 4–100/4.0 column (Metrohm AG, Herisau, Switzerland). The detection limits (µeq L^−^¹) were 2.29 (SO_4_^2−^), 1.77 (NO_3_^−^), 2.26 (Cl^−^), 2.50 (Ca^2+^), 3.29 (Mg^2^⁺), 1.79 (K⁺), 2.22 (NH_4_⁺), and 1.74 (Na⁺). 

Data quality assurance was monitored using two criteria: electric conductivity and anion/cation ratio. In the first one, the Ion Sum is considered: if total concentration is lower than 100 µeq L^−^¹, Ionic Difference is considered (the difference between anions and cations). In the second criteria, theoretical EC is compared with measured EC and the acceptability criteria depends on the measured Ion Sum [54].

This technique was used to analyze the rain samples collected (“Analysis of wet atmospheric deposition”) and the lixiviate samples (“Accelerated weathering”). Although this technique is not as often used as others (XRD, XRF, MO), it is very powerful to monitor decay through the analysis of wet atmospheric deposition and the lixiviates. A difference in ion concentration between them means that there is either the dissolution of some compounds inside the stone or that some ions still remain inside the porous network, implying salt crystallization or ion exchange; still, in the available literature where IC was used, there is little discussion on the matter (i.e., [55]).

### 2.5. Statistical Analysis

A Shapiro–Wilk test was performed on mass, WAC and OP results to verify if they follow a normal distribution. This test is recommended for a small number of samples [56]. A standardized paired *t*-test was used to confirm significant differences before and after the accelerated weathering treatments [57], and confirm if mass, OP and WAC are significantly different after the accelerated weathering test. A Friedman two-way analysis of variance by ranks was used, which “tests the null hypothesis that the k-repeated measures or matched groups come from the same population or populations with the same median” [58]. This test was done to ensure that there were no significant differences in chemical composition for the different artificial rain batches, the objective composition nor among all seven. A Wilcoxon signed-rank test was used “to compare two related samples, matched samples, or to conduct a paired difference test of repeated measurements on a single sample to assess whether their population mean ranks differ” [59]. This test was done to analyze if there was a significant difference in chemical composition between the irrigated solution and the samples taken after the solution interacted with red basalt.

## 3. Results and Discussion

Results of this study are presented in four categories: analysis of wet atmospheric deposition (sampled rain), water absorption properties, mass difference and ion chromatography used in the analyses of accelerated weathering test samples.

### 3.1. Analysis of Wet Atmospheric Deposition

Annual average data from meteorology (obtained from SIMAT) is summarized in Table 2. Temperature (T), relative humidity (RH), wind speed (WS) and wind direction (WD) correspond to Merced (MER) station, whereas precipitation corresponds to MCM station. T, RH and wind conditions are not extreme, so it is expected that decay does not depend mainly on them [28].

During the 2014–2019 period, 5858 mm of accumulated precipitation were measured in 152 samples at MCM station and were classified depending on their pH as acidic (pH < 5.6) or non-acidic (pH > 5.6), as shown in Table 3. Of this total, 22% of the volume corresponded to acidic precipitation but represented only 16% of the samples. This volume, although apparently low, can still play an important role in basalt decay, because plagioclase dissolution is higher in both low and high pH values, thus facilitating cation exchange. On average, each year precipitates about 976 mm of total rain (213 mm as acidic rain and 763 mm of non-acidic rain).

Table 3 shows no acid rain for 2017. MCM station is placed at NE Mexico City (low altitude and in a plain region), which presents intermediate precipitation rates and the highest neutralization indices, with fractional acidity between 2% and 4% (neutralization can be up to 98%), so mainly no acid rain is identified. This neutralization is probably due to the presence of particulate matter (PM) containing Ca²⁺, Mg²⁺, Na⁺, and K⁺, from commercial and vehicle traffic activity, or because it is downwind from the industrial area located to the north of the MCM station. The results presented on Table 3 show why it is important to develop damage functions relating decay to atmospheric and not to temporal parameters: the volume of rain. Although it can be summarized as an arithmetic mean and has a relatively large standard deviation, simply because there are some drought periods. For example, the global volume sampled during 2014 is almost thrice the global volume sampled during 2017.

The VWM for each kind of deposition of the 152 samples collected in the five years period (Appendix A) was calculated using Equation (2), for each ion. Results are summarized in Table 4.

### 3.2. Water Absorption Properties

One of the objectives of this research was to find a different way to monitor decay, because although the mass loss is still the most widely used quantification of decay (even if they are expressed as “surface recession”, as in [8,13]), and is undoubtedly a good approach. However, it is not normally possible to measure mass on the actual building. Instead, WAC and OP appear to be a suitable alternative to monitor decay in a non-destructive way. Results of WAC and OP before and after the accelerated weathering test are presented in Figure 5 and Table 5.

Results show that OP and WAC decreased in global deposition, but these variables increased during both acid and no-acid treatments (Figure 5). Theoretically, OP and WAC would increase as the material deteriorates because of the depletion of the minerals of the material, the washing of recrystallized salts, clay minerals and decay products [60].

There are several reasons why OP and WAC diminish after the accelerated weathering. First, the volume irrigated to global group was higher than acid and no-acid cases, and it can be the main cause of the difference because water produces all the decaying mechanisms. Second, the pore network seems to be closing, mainly due to the washing of insoluble compounds, such as clay minerals [60] that obstruct the pore network, as can be seen in Figure 3b, in which non-soluble particles were captured in the filters. The decrease in OP and WAC cannot be produced by salt crystallization because the crystallized salts are soluble, and after the WAC test, salts were removed. Finally, although precipitation volume is important, the chemical composition also plays an important role. It will be discussed in the section “Ion Chromatography used in analysis of accelerated weathering” that pH is indeed an important variable affecting decay. Future research is needed to distinguish if changes are due to a higher ion concentration or because of pH, using for example, the salinity as an indirect measure of rain “cleanliness” in the future.

All three batches’ mass, WAC and OP follow a normal distribution (Shapiro–Wilk test, Table 6). In all three cases, the mass difference for all six specimens was significant when a standardized paired t-test was run (Table 6). OP and WAC were significantly different for global and acid precipitation types, but not for no-acid precipitation. Hence, there is no evidence that no-acid wet atmospheric deposition influences water absorption properties, even if the mass has a significant difference. That is, a less acid solution does not promote solubilization of minerals nor washing of clay materials because the EC does not change significantly, as will be further discussed. For this class of wet atmospheric deposition, the main decay mechanisms are probably because of the crystallization of soluble ions inside the materials as a salt, or cation exchange, but XRD and XRF analysis are needed. However, since in the global and acid precipitation classes, WAC and OP have a significant difference, they can be cautiously used as an in situ decay monitoring variable, although mass is still the best one that can be used. For OP and WAC, only general trends are presented because the specimens’ porosity variability was high.

### 3.3. Mass Difference

Tezontle is a very porous material; its open porosity can be almost 10% (Table 5). Although the specimen dimensions are similar, the difference in mass is large, which results from the genesis of the material. Hence, the mass difference after the accelerated weathering is almost twice for some specimens. To reduce this variability, results are presented as percentage mass loss compared with initial mass in Figure 6, Figure 7 and Figure 8. However, in the future, when a materials’ porosity is heterogeneous, different specimen dimensions may be considered. In all three figures, the main difference in mass is due to loss of absorption capacity, because of the washing of insoluble particles (all the filters retained reddish particles as can be seen in Figure 3b), probably clay minerals. If the mass losses were produced mainly by lixiviation of the tezontle’s matrix, dry and wet masses would be similar. Despite previous publications (i.e., [8,11,13,52]), mass loss does not behave linearly: there are volumes at which mass increases and others in which they decrease. Behavior can be presented as a fifth-degree polynomial function for the simulated period, correlating the mass loss with the irrigated volume per group of wet atmospheric deposition.

Three specimens were selected (one per class of deposition: G3, N4, A6) to exemplify the behavior of tezontle under the different chemical composition irrigation. The functions that better adjusted to their behavior are shown in Figure 7 and Table 7, which can be adjusted with a fifth-order equation (Equation (3)).

V represents the volume of accumulated rain in L, and md represents the mass difference in grams. The coefficients can have a negative symbol and they would depend on the behavior of the specimen, as can be seen in Table 7. It is important to highlight that in Figure 7, decay is presented both as a function of time (Figure 7a) and precipitation (Figure 7b) to better exemplify the reason to prefer the latter: the volume of rain that will fall in outdoor exposure for 10 years will be around 2200 mm for acid rain, but 12,000 mm for global precipitation.
(3)$$F\left(V\right)=m_{d}={aV}^{5}+{bV}^{4}+cV^{3}+dV^{2}+eV+g$$

Dry masses were also measured, but only at the beginning and the end (Figure 8, Table 7), to compute a simpler function relating volume of rain with mass change (specimens G2, N4, A4). Representing decay as the difference in mass between the beginning and the end of the accelerated test is the easier way, because to relate two points, a straight line with good correlation can be computed, although it does not necessarily capture the complexity of decay, especially on highly porous materials such as tezontle.

Figure 6 and Figure 8 show that the mass difference depends on the volume of rain, and that they all have a cyclic behavior. Changes are more pronounced for the acid class, meaning greater difference in mass probably because of lixiviation of minerals and washing of insoluble compounds (such as clay minerals). This can be explained because one of the most abundant minerals in tezontle is plagioclase [47], a series of solid solutions of sodium and calcium aluminosilicates, whose dissolution has a minimum of around 5.7 [61]. On the other hand, for the no-acid class the main difference is probably due to only washing of clay minerals. Global class behaves differently, because to its pH, there is not a clear mechanism controlling decay, and probably they both compete provoking a different coefficient sign modelling the behavior, and they decay relatively more slowly (Table 7, Figure 6 and Figure 8).

If the equations from Table 7 are evaluated, acid rain would produce a greater effect on the red basalt than no acid and global rain types, although their volume is considerably lower. Therefore, acid rain effect is greater for the same volume than no acid and global deposition. However, even if pH is a critical variable, decay depends mostly on the volume of each rain class. Further research is needed, because chemical composition is much more complex than just pH, and some specific phenomena can happen, such as common ion effect, cation exchange and dissolution of specific compounds.

### 3.4. Ion Chromatography Used in Analysis of Accelerated Weathering Test Samples

Applying the Friedman two-way analysis of variance by ranks test to the artificial rain samples (and using the Bonferroni correction), it was shown that there was only a significant difference between batches TM-N0-L-B and TM-N0-L-6. When compared with the objective composition and the rest of the batches, the result was the same as the rest of the comparisons. In summary, there was no evidence of a significant difference in chemical composition.

The results of the Wilcoxon signed-ranked test, which compares the concentration of each ion in the artificial rain with concentration of each ion in lixiviation samples, are presented in Table 8. A null hypothesis means that there is no evidence of significant difference in composition overall, whereas an alternative hypothesis means that there is evidence of significant difference between composition of the lixiviate and the artificial solution irrigated.

The concentration of ions increases after the interaction of the artificial rain solution with the building material, but in most cases, there is not enough evidence to conclude this. The only general trend is that the concentration of calcium ion (Ca^2+^), increases when compared with blanks (Figure 9). The first possible source of cations in all solutions, especially in artificial acid precipitation is lixiviation, or dissolution, of the main minerals in the basalt, especially Ca^2+^ (rocks tend to be basic in the region [62] so they have more calcium than sodium), although plagioclases dissolution is easier in basalt for acidic and non-acidic conditions [61]. In fact, Ca^2+^ availability also produces pH neutralization (Figure 9). This change is observable in all three sets of data, but is more evident in the acid one, because the dissolution of plagioclases is easier in those conditions [61].

One important meaning of this experimentation is the fact that when simulating material decay, using a catastrophic scenario is not the best. It will cause not worse, but different decay that might not represent the complexity of the problem. No acid deposition is the one which produces a lower mass difference in the material, and theoretically it would produce more decay because of Ca availability [37]. Global deposition behaves differently, so at that pH it is not a clear decay-controlling mechanism. There are present at least three reasons. First, the dissolution of minerals because in some cases, ion concentration increases. Second, the washing of insoluble compounds as is observable in Figure 2. Third, salt crystallization because in some cases, ion concentration reduces.

Acid rain deposition is the one which produces a greater decay in the material, probably due to lixiviation of minerals as shown by the enrichment of the solution, but leaching of cations is also possible [63].

There are two indirect ways of measuring the concentration of compounds in a solution: EC and salinity (a term used for referring the total amount of soluble ions present in a solution). When comparing leached samples among themselves, an increase in EC would mean a significant dissolution of compounds of the specimens. It can be seen in Figure 9d–f that EC did not change significantly in the samples of acid and no-acid deposition; however, there is a significant mass change, suggesting the loss of non-soluble compounds, most probably clay minerals. This was not the case for global deposition, but since at those conditions, solubility of plagioclases is minimum, there is another explanation for cation enrichment: cation exchange.

Cation exchange, especially for clays, is such an important enrichment mechanism that it is commonly measured in agriculture [64]: the ions of the solution are interchanged with ions in the clay, and generally NH_4_^+^ in solution substitutes Ca^2+^, Na⁺, K⁺, Mg^2+^ in that order. Results from Table 9 and Appendix B show that whereas NH_4_^+^ concentration diminishes, Ca^2+^ and K^+^ concentration increase. However, this phenomenon is not responsible for significant changes in OP for global and acid batches [65].

After the Spearman test was performed (Appendix C), a strong correlation was found to exist between calcium and potassium, bicarbonate and chloride. The presence of potassium probably results from the K-feldspars of the material, whereas bicarbonate and chloride are present because there were salts inside the porous of the material that were dissolved and washed away, but the specific allotropes that produce decay are a topic to be researched in the future. However, it must be considered that tezontle is a basalt, commonly richer in Ca^2+^ than Na⁺, and that is probably the main source of calcium. We would expect, eventually, the formation of decay crust under natural conditions, because as it has been published, the availability of Ca^2+^ is related to crust formation on volcanic stone [37].

In the global artificial precipitation, there is evidence of salt recrystallization because Cl^−^, K^+^, NH_4_^+^ and HCO_3_^−^ concentrations decrease and increase cyclically. This, along with clay washing, can also explain the decrease of WAC and OP.

In no-acid precipitation, there is only a decrease in the first lixiviation, and then, ions are depleted from the building material whereas in the artificial acid rain, the concentration of ions only increases. This means that simulating only a pH type of rain only works well when the main decay mechanism is lixiviation, because the common ion effect reduces the probability of formation of salts.

Finally, and since the main decay mechanism seems to be caused by washing of insoluble compounds (probably clay minerals), it would be important to research the mechanical effect they produce, because in some cases, the volumetric strain produced during weathering can be more important than chemical depletion [66].

## 4. Conclusions

An accelerated weathering test showed that tezontle, a red basalt commonly used as building material in central Mexico, suffers from decay due to wet atmospheric deposition. The methodology employed is suitable for monitoring decay, although changing specimen dimensions can be changed to diminish the effect of porosity heterogeneity. There was a significant mass loss, caused probably by the washing of non-soluble compounds (most likely clay minerals), dissolution of several compounds (especially Ca-bearing minerals that need to be researched deeper in the future), salt crystallization and cation exchange.Water Absorption Capacity and Open Porosity were tested, and although mass difference showed better results, these can be cautiously employed as two in situ variables for monitoring decay when no other test is available.Basalt decay was modeled for a period of ten years, and a first approach to decay equations was presented as fifth-degree functions. The main source of decay in the rain is the acidic events, in which mass loss is higher than non-acidic events. However, precipitation produces mass loss regardless of rain pH and volume is indeed the critical variable in wet atmospheric deposition.Decay of basalt is not linear, and when modeling it, the use of measurable variables, such as rain volume, instead of arbitrary units (such as years) can help to better forecast decay rates of the building material.Volcanic-stone-based heritage is present worldwide, but knowledge on their decay mechanisms and the variables controlling them needs to be enhanced further. Statistical analysis of stone properties is important to model the decay of basalt; however, different methodologies with different time spares, local chemical compositions and precipitation volume should be researched for better forecast.

## Figures and Tables

**Figure 1 materials-16-05644-f001:**
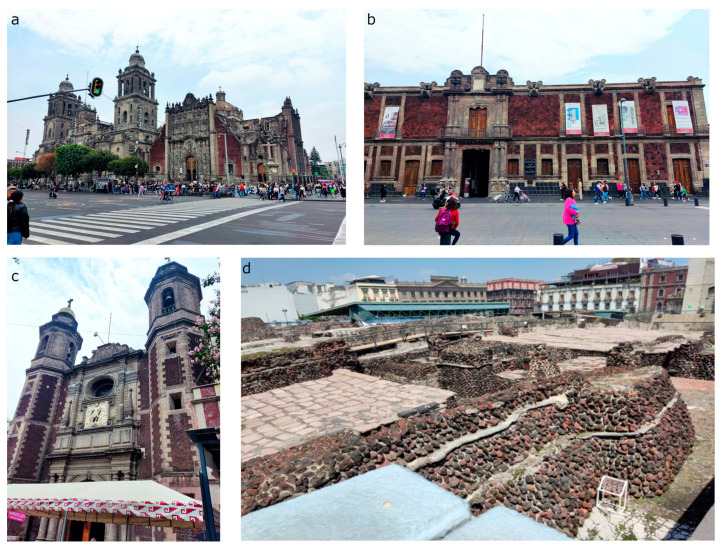
Monuments built with red basalt, tezontle: (**a**) Mexico City Metropolitan Cathedral façade (**b**) Palace of the Counts of Santiago de Calimaya (**c**) Saint Michael Archangel church (**d**) Templo Mayor.

**Figure 2 materials-16-05644-f002:**
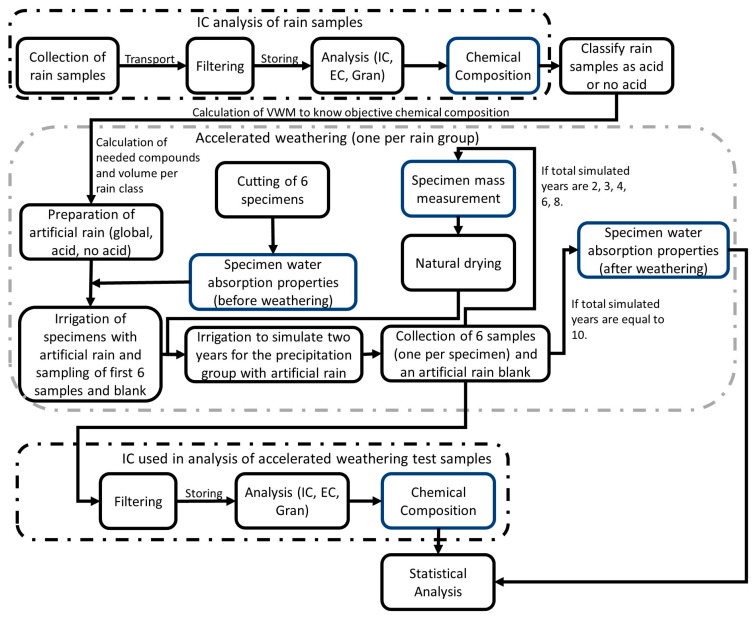
Scheme of the complete experimentation. Each dotted line represents a section of the experimentation. The blue blocks mean a result, whereas black block represent an experimentation cycle performed.

**Figure 3 materials-16-05644-f003:**
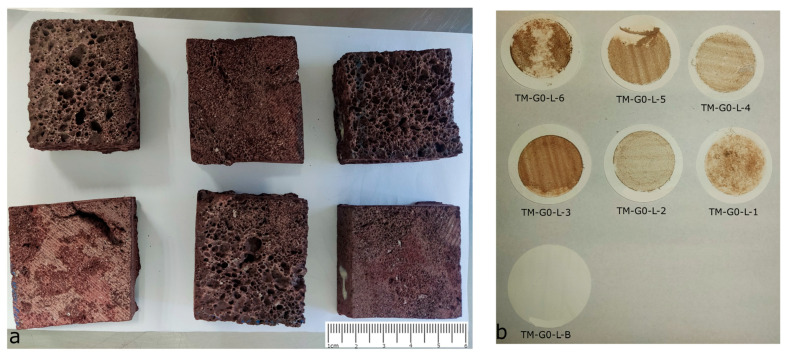
(**a**) Red basalt specimens for accelerated weathering. Scale is in centimeters (**b**) Insoluble compounds visible after filtration of the mentioned samples.

**Figure 4 materials-16-05644-f004:**
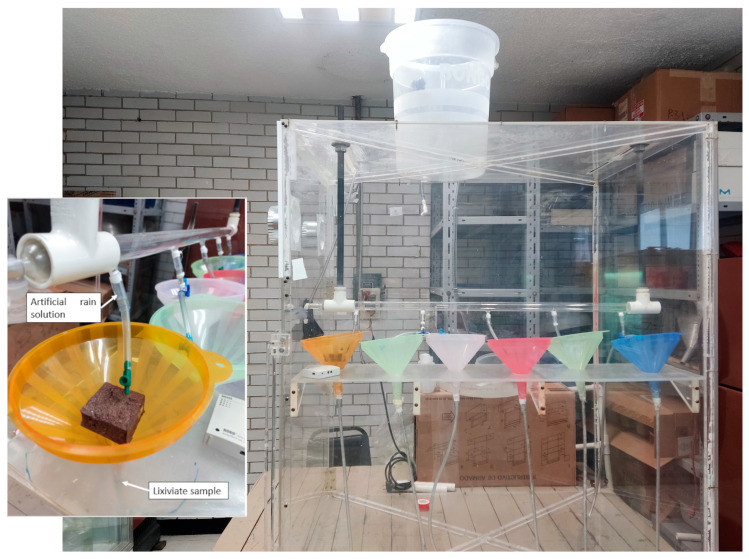
Accelerated weathering chamber used in this study.

**Figure 5 materials-16-05644-f005:**
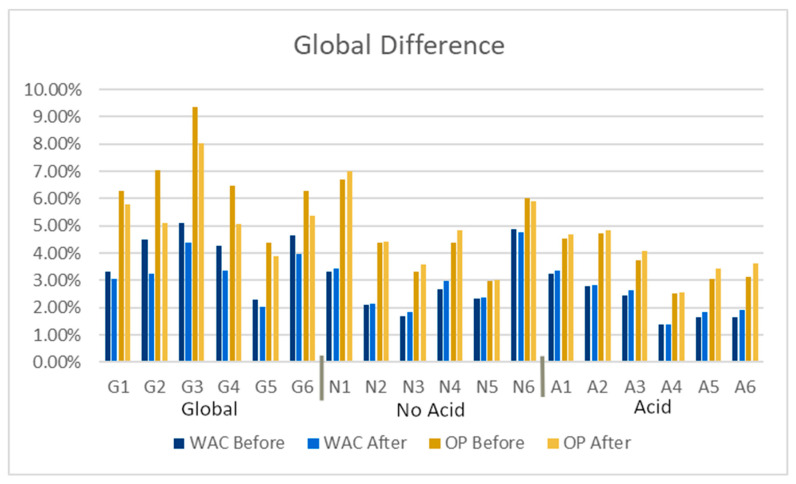
Comparison of %WAC and %OP before and after the test for each specimen.

**Figure 6 materials-16-05644-f006:**
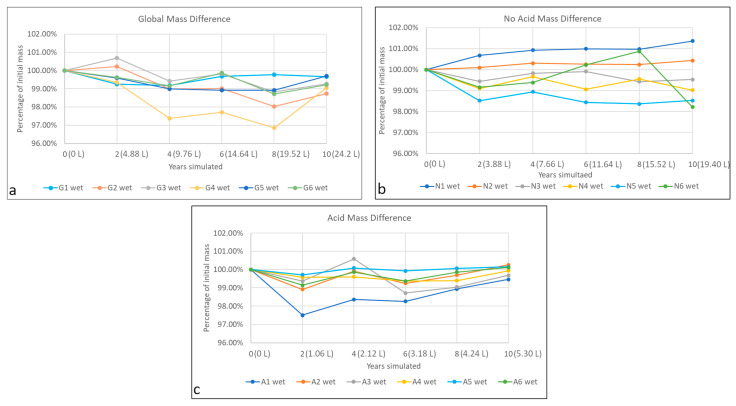
Mass evolution per group of rain. In parentheses, the accumulated artificial rain irrigated per specimen. Each line represents the change of partially wet mass for a single specimen: (**a**) Global mass difference, (**b**) No acid mass difference, (**c**) Acid mass difference.

**Figure 7 materials-16-05644-f007:**
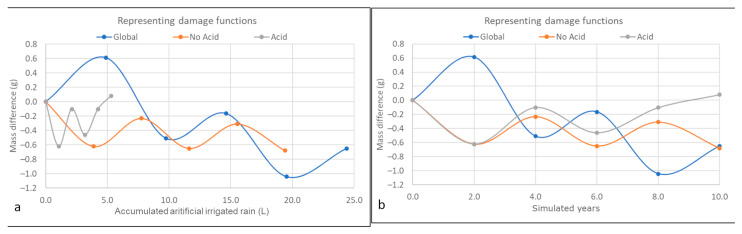
Mass evolution as function of accumulated artificial rain irrigated.

**Figure 8 materials-16-05644-f008:**
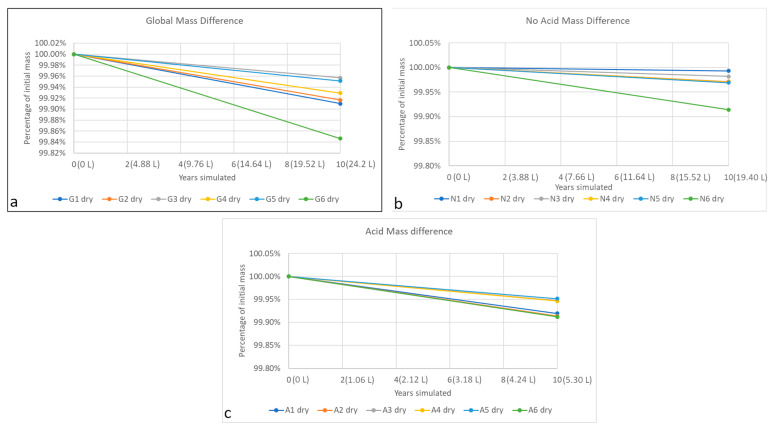
Mass evolution per class of rain. In parentheses, the accumulated artificial rain irrigated per specimen. Each line represents the change of dry mass for a single specimen: (**a**) Global mass difference, (**b**) No acid mass difference, (**c**) Acid mass difference.

**Figure 9 materials-16-05644-f009:**
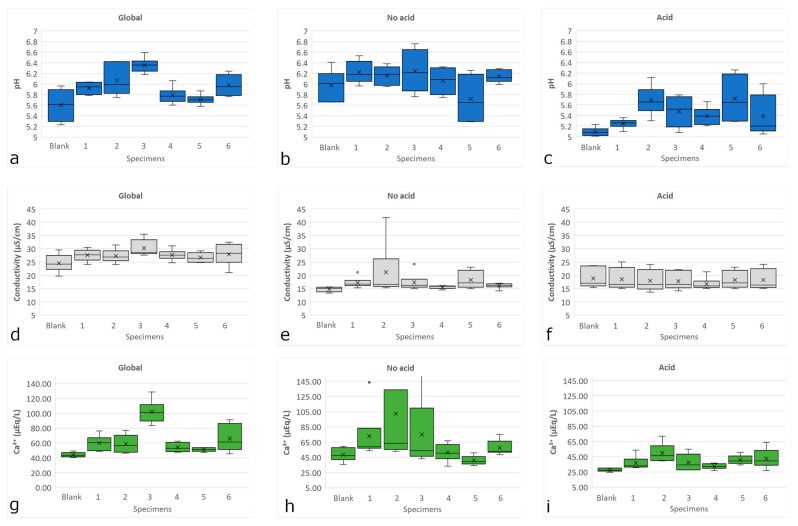
Change of pH (**a**–**c**), EC (**d**–**f**) and Ca^2+^ concentration (**g**–**i**) for each specimen. Plots correspond to global (**a**,**d**,**g**), no acid (**b**,**e**,**h**) and acid (**c**,**f**,**i**) artificial rain. The × represents the average value; horizontal lines inside the box represent the median; bottom and top of the boxes represent the 25 and 75% percentiles, respectively; and the bottom and the top whiskers represent the 5 and 95% percentiles, respectively.

**Table 1 materials-16-05644-t001:** Compounds used for artificial rain solutions.

Global			Acid			No Acid		
Volume of sampled rain (mm)		976.30	Volume of sampled rain (mm)		213.64	Volume of sampled rain (mm)		762.66
Annual volume per specimen (L)		2.44	Annual volume per specimen (L)		0.53	Annual volume per specimen (L)		1.91
Total Volume (L)		147.00	Total Volume (L)		33.00	Total Volume (L)		116.00
pH		5.5	pH		4.9	pH		6.3
Compound	Mass (g)	Volume (mL)	Compound	Mass (g)	Volume (mL)	Compound	Mass (g)	Volume (mL)
NH_4_NO_3_	0.5217		Ca(NO_3_)_2_	0.4583		CaCO_3_	0.9205	
CaSO_4_	1.8231		NaNO_3_	0.0177		Ca(NO_3_)_2_	0.2657	
MgCl_2_	0.0963		Mg(NO_3_)_2_	0.0442		CaCl_2_	0.0772	
NaCl	0.0433		HCl	0.0208		MgCl_2_	0.0769	
K_2_CO_3_	0.1032		KCl	0.0040		KCl	0.0143	
H_2_SO_4_ (18.4 M)	0.6879	0.378	H_2_SO_4_	0.0087		KNO_3_	0.0149	
NH_4_OH (14 M)	0.2458	0.061	NH_4_NO_3_	0.0002		NaNO_2_	0.0098	
			H_2_SO_4_ (18.4 M)	0.7259	0.399	H_2_SO_4_ (18.4 M)	1.4365	0.789
			NH_4_OH (14 M)	0.2593	0.065	NH_4_OH (14 M)	0.5132	0.128

**Table 2 materials-16-05644-t002:** Annual average meteorological conditions of Merced Meteorological Station. Averages were calculated from hourly data reported by SIMAT.

Year	T _average_ (°C)	RH (%)	Wind Speed (m/s)	Wind Direction (°)
2014	17.3	52.0	2.1	326.3
2015	17.2	58.5	2.1	326.3
2016	17.3	54.9	2.2	326.3
2017	17.6	52.3	2.1	326.3
2018	17.5	55.6	2.1	326.3
2019	18.5	51.1	2.1	326.3
Mean	17.6	54.1	2.1	326.3 (NW)

**Table 3 materials-16-05644-t003:** Wet atmospheric deposition sampled and classified by pH for the 2014–2019 period for the MCM station.

Group of Wet Atmospheric Deposition	Year	2014	2015	2016	2017	2018	2019	Total
Global	Samples	37	22	24	20	24	25	152
	Volume (mm)	1965	710	774	686	768	955	5858
No Acid	Samples	10	6	3	0	3	2	24
	Volume (mm)	809	203	103	0	68	99	1282
Acid	Samples	27	16	21	20	21	23	128
	Volume (mm)	1157	508	670	686	700	856	4576

**Table 4 materials-16-05644-t004:** Volume weighted mean composition per class of wet atmospheric deposition.

Group of Wet Atmospheric Deposition	Global	Acid	No Acid
Ion	x_w_(μEq/L)	x_w_ (μEq/L)	x_w_ (μEq/L)
[H^+^]	4	0	4
[Na^+^]	5	3	3
[NH_4_^+^]	98	42	56
[K^+^]	3	1	1
[Mg^2+^]	3	2	1
[Ca^2+^]	45	25	19
[F^−^]	1	0	0
[Cl^−^]	10	3	7
[NO_3_^−^]	45	19	27
[SO_4_^2−^]	69	26	43
[HCO_3_^−^]	29	22	6
[NO_2_^−^]	1	0	0
Weighted pH	5.5	4.9	6.6
Uncertainty: ±0.56 μEq/L for Cl^−^; ±0.32 μEq/L for NO_3_^−^; ±0.42 μEq/L for SO_4_^2−^; ±0.51 μEq/L for K^+^; ±0.87 μEq/L for Na⁺, ±1.11 μEq/L for NH_4_^+^; ±1.65 μEq/L for Mg^2+^; ±1.00 μEq/L for Ca^2+^.			

**Table 5 materials-16-05644-t005:** Mass, WAC, and OP averages and standard deviation before and after the accelerated weathering test.

Before Treatment										
Deposition Group	Saturation Mass (g)		Dry Mass (g)		Vop (cm³)		%OP		%WAC	
	Average	Std. Deviation	Average	Std. Deviation	Average	Std. Deviation	Average	Std. Deviation	Average	Std. Deviation
Global	79.4481	12.2473	76.4481	13.3462	3.0000	0.7538	6.63%	1.61%	4.01%	1.04%
No Acid	80.9924	19.4507	78.8626	21.1188	2.1298	0.7080	4.63%	1.47%	2.83%	1.14%
Acid	67.7868	10.3265	66.3994	11.5237	1.3874	0.2749	3.61%	0.88%	2.19%	0.74%
After treatment										
Deposition group	Saturation mass (g)		Dry mass (g)		Vop (cm³)		%OP		%WAC	
	Average	Std. Deviation	Average	Std. Deviation	Average	Std. Deviation	Average	Std. Deviation	Average	Std. Deviation
Global	78.8962	13.5179	76.3889	13.3556	2.5073	0.6480	5.54%	1.38%	3.34%	0.82%
No Acid	81.0465	21.3526	78.8395	21.1252	2.2071	0.7062	4.80%	1.47%	2.92%	1.08%
Acid	67.8375	11.3111	66.3547	11.5215	1.4828	0.2568	3.86%	0.85%	2.34%	0.74%

**Table 6 materials-16-05644-t006:** Shapiro–Wilk and *t*-Student’s test for standardized paired data.

Mass					
	Deposition group	W	*p*	Distribution	
Shapiro–Wilk test	Global	0.9199	0.5050	Normal	
	No Acid	0.9074	0.4193	Normal	
	Acid	0.9562	0.7897	Normal	
	Deposition group	t	*p*	Average	¿Significant difference?
*t*-Student’s test	Global	−6.9910	0.0009	−0.0592	Yes
	No Acid	−4.0123	0.0102	−0.0232	Yes
	Acid	−9.6010	0.0002	−0.0447	Yes
Water Absorption Capacity (WAC)					
		W	*p*	Distribution	
Shapiro–Wilk test	Global	0.9331	0.6045	Normal	
	No Acid	0.9652	0.8585	Normal	
	Acid	0.9322	0.5972	Normal	
		t	*p*	Average	¿Significant difference?
*t*-Student’s test	Global	−4.3033	0.0077	−0.007	Yes
	No Acid	1.7229	0.1455	0.0009	No
	Acid	3.5813	0.0159	0.0015	Yes
Open Porosity (OP)					
	Deposition group	W	*p*	Distribution	
Shapiro–Wilk test	Global	0.9218	0.5181	Normal	
	No Acid	0.9586	0.8090	Normal	
	Acid	0.9516	0.7532	Normal	
	Deposition group	t	*p*	Average	¿Significant difference?
t-Student’s test	Global	−4.7085	0.0053	−0.0110	Yes
	No Acid	1.9707	0.1058	0.0017	No
	Acid	3.2627	0.0224	0.0025	Yes

**Table 7 materials-16-05644-t007:** Equations representing wet and dry mass difference per group of deposition.

Group of Wet Atmospheric Deposition	Dry Mass Difference	Wet Mass Difference
Global	$m_{d}=-0.0025V$	$m_{d}{=0.000003V}^{5}-0{.0021V}^{4}+0.0451V^{3}-0.4116V^{2}+1.2802V-7{\cdot 10}^{-11}$
No Acid	$m_{d}=-0.0011V$	$m_{d}{=-0.00006V}^{5}+{0.003V}^{4}-0.0517V^{3}+0.3744V^{2}-0.9951V-4{\cdot 10}^{-11}$
Acid	$m_{d}=-0.0085V$	$m_{d}{=-0.038V}^{5}+0{.5218V}^{4}-2.5344V^{3}+5.1415V^{2}-3.7622V+2{\cdot 10}^{-11}$

*V* represents the volume of precipitation of each type (L), and *m_d_* represents the mass difference (g).

**Table 8 materials-16-05644-t008:** Significant differences between blanks and objective composition, and samples and blanks.

Years of Precipitation Simulated	Blank	G1	G2	G3	G4	G5	G6
0	N	A	N	A	A	N	N
2	N	N	A	A	N	N	N
4	N	A	N	A	A	A	A
6	N	N	N	N	N	N	N
8	N	A	A	A	A	N	A
10	N	A	A	A	N	N	A
Years of precipitation simulated	Blanks	N1	N2	N3	N4	N5	N6
0	N	N	A	A	A	N	N
2	N	A	N	N	N	N	N
4	N	A	N	N	N	N	N
6	A	A	A	A	A	A	A
8	N	N	N	N	N	N	N
10	N	A	A	N	N	N	N
Years of precipitation simulated	Blanks	A1	A2	A3	A4	A5	A6
0	N	A	A	N	N	N	A
2	A	N	N	N	N	N	N
4	N	N	N	N	N	N	N
6	A	N	N	N	N	A	A
8	N	N	N	N	N	N	N
10	N	N	A	N	N	N	N

N stands for “Null Hypothesis”. A stands for “Alternative Hypothesis”.

**Table 9 materials-16-05644-t009:** Summary of statistical tests.

Deposition Group	Years of Accelerated Weathering	¿Significant Difference?	Samples	Ion Concentration Change
Global	0	Alternative hypothesis	G1, G3, G4	Increased all ions, except NH_4_^+^, which diminishes. Na^+^ doubled
		Null hypothesis	G2, G5, G6	
	2	Alternative hypothesis	G2, G3	Increased all ions, especially Ca^2+^ and HCO₃^−^
		Null hypothesis	G1, G4, G5, G6	Slight increase in Ca^2+^, Cl^−^
	4	Alternative hypothesis	G1, G3, G4, G5, G6	Increase Ca^2+^, NH_4_^+^, SO_4_^2−^, HCO₃^−^
		Null hypothesis	G2	Slight increase in concentration overall
	6	Alternative hypothesis	0	-
		Null hypothesis	G1, G2, G3, G4, G5, G6	Slight increase in NH_4_^+^, Ca^2+^, NO_3_^−^, SO_4_^2−^ reduced K^+^, Cl^−^, HCO_3_^−^
	8	Alternative hypothesis	G1, G2, G3, G4, G6	Slight increase NH_4_^+^, Mg^2+^, Cl^−^. Large increase NO_3_^−^, SO_4_^2−^, Ca²⁺
		Null hypothesis	G5	Slight increase NH_4_^+^, Ca^2+^, HCO_3_^−^ large increase NO_3_^−^, SO_4_^2−^, decrease Cl^−^
	10	Alternative hypothesis	G1, G2, G3, G6	Slight increase in all ions
		Null hypothesis	G4, G5	A slight decrease in SO_4_^2−^, increase in the rest of the ions
No Acid	0	Alternative hypothesis	N1	Slight decrease in K^+^, NH_4_^+^, Mg^2+^. Increase Ca²⁺, HCO_3_^−^
		Null hypothesis	N2, N3, N4, N5, N6	Increase K^+^, Ca^2+^, Cl^−^, NO_3_^−^, SO_4_^2−^, HCO_3_^−^. Decrease NH_4_^+^
	2	Alternative hypothesis	N1	Increase overall, especially in K^+^, HCO₃^−^
		Null hypothesis	N2, N3, N4, N5, N6	General increase, especially higher in Ca^2+^, K^+^, HCO_3_^−^
	4	Alternative hypothesis	N1	Slight increase overall, especially higher in HCO_3_^−^
		Null hypothesis	N2, N3, N4, N5, N6	Slight decrease NO_3_^−^, SO_4_^2−^. Mild increase in the rest.
	6	Alternative hypothesis	N6	Slight increase overall, especially higher in Ca^2+^, HCO_3_^−^
		Null hypothesis	0	-
	8	Alternative hypothesis	0	-
		Null hypothesis	N6	Slight decrease in Cl^−^, amild increase in Ca^2+^
	10	Alternative hypothesis	N1, N2	Slight increase Ca^2+^
		Null hypothesis	N3, N4, N5, N6	Slight increase Ca^2+^
Acid	0	Alternative hypothesis	A1, A2, A6	Increase Na^+^, K^+^, Cl^−^, SO_4_^2−^. Ca^2+^ concentration duplicated. Slight decrease NH_4_^+^
		Null hypothesis	A3, A4, A5	Slight increase in K^+^, Ca^2+^, Cl^−^
	2	Alternative hypothesis	0	-
		Null hypothesis	A1, A2, A3, A4, A5, A6	Slight increase in K^+^, Ca^2+^, Cl^−^
	4	Alternative hypothesis	0	-
		Null hypothesis	A1, A2, A3, A4, A5, A6	Not a clear trend
	6	Alternative hypothesis	A5, A6	Slight increase in NO_3_^−^, HCO_3_^−^. Na^+^, K^+^, Ca^2+^, Cl^−^ more than doubled
		Null hypothesis	A1, A2, A3, A4	Mild increase Na^+^, K^+^, Ca²^+^, Cl^−^, NO_3_^−^. Slight decrease overall.
	8	Alternative hypothesis	-	
		Null hypothesis	A1, A2, A3, A4, A5, A6	Mild decrease Na⁺, NH_4_^+^. Slight increase Ca^2+^, Cl^−^
	10	Alternative hypothesis	A2	Increase Na^+^, K^+^. Concentrations of Ca^2+^, Cl^−^ more than doubled
		Null hypothesis	A1, A3, A4, A5, A6	Not a clear trend

## Data Availability

Supporting data can be found in the Appendix A, Appendix B and Appendix C, but wet atmospheric deposition data and meteorological data was obtained from the SIMAT, available at: http://www.aire.cdmx.gob.mx/default.php, accessed on 20 June 2023.

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
