# Peer review of "Quantifying Decay Due to Wet Atmospheric Deposition on Basalt"

_materials, 2023, doi:10.3390/ma16165644_

Round 1

Reviewer 1 Report

Dear authors the paper entitled Quantifying decay due to wet atmospheric deposition on basalt is quite interesting as dealing with possible testing construction materials in the future in outdoor conditions. The work deals with laboratory tests but the results are promising in sustaining additional applications in outdoor. I appreciate the effort in trying to quantify the decay and, in particular, the attention given to address other parameters than the common weight loss for stone decay.

English can be improved, too often you write long sentences that are hard to follow. I would kindly invite you to revise the text and pay attention also to typos. Caption of tables do need to be consistent as well as format text in table and figure.

Please see my comments that are also reported in the text as highlighted

Page 2 line 57-58 please separate in two sentences as too many facts are put together, the sentence structure in unclear.

Page 2 line 57-58 please separate in two sentences as too many facts are put together, the sentence structure in unclear. The two main concepts will be more clear and effective if divided in two sentences.

Page 2 line 80 correct the capital letter line 89 correct That with This is done..

Page 5 line 167 please change…this limiting can be overcome….with…this limit/aspect can be overcome

Page 6 line 192-193 please correct the sentence

Page 6 line 195-200 again too long and heavy sentence. Make a full stop after ..samples collected ,monthly.

Page 7 line 220 please use if possible always the same concentration either N or M but do not mix it.

Page 8 line 261 – 262 something is not working….and confirm is (maybe you wanted to write IF?) mass, OP and….

Page 8 line 266-267 please correct the sentence.  This test was done to ensure that there were..

Page 8 line 284 check the size and format of table caption , be consistent . See also table 3 and 4 – correct the font and size

Page 10 Figure 4 authors need to have the same maximum unit for all the conditions. It is difficult to get the differences among them if you have different scales.

Page 11 line 328 check the caption of table 5

Page 11 line 332-333 have a look to the sentence, it is not clear . Line 333-337 again too long and unclear discussion. You need to separate the different comments and help the future reader to understand.

Page 14 Figure 6 Can you please explain me why here you have data for acid up to 5 on the X axis while in Figure 7 we do have all different scales on X axis? Honestly, I do not get how you can compare data or better behaviour among situation if you stop at different points

English is medium but few sentences need to be revised and made more structured. The authors often used too long and complicated sentences.

Typos are also present but, in general, it is ok.

Author Response

Reviewer 1

We appreciate all your comments and suggestions because following them improved the manuscript.

English can be improved, too often you write long sentences that are hard to follow. I would kindly invite you to revise the text and pay attention also to typos. Caption of tables do need to be consistent as well as format text in table and figure.

Besides the changes in the suggested lines, some other sentences were shortened for clarity.

Page 2 line 57-58 please separate in two sentences as too many facts are put together, the sentence structure in unclear. The two main concepts will be more clear and effective if divided in two sentences.

We agree that the sentence would be clearer if divided. Hence, the sentence: “Although all building materials need to be studied to ensure the preservation of building heritage worldwide, this research was done to increase knowledge on volcanic stone-built heritage, discussing the effect that wet atmospheric deposition has on its de-cay, as just preliminary results of a multidimensional research involving material nature, meteorology, and air quality aspects of the damage it can suffer.”

Was changed for:

To ensure preservation of built heritage worldwide, all building materials need to be studied. This research was done specifically to increase knowledge on volcanic stone-built heritage. Here, the effect of wet atmospheric deposition on decay is discussed, but it is only a first approach: in the future, multidimensional research also involving meteorology, material nature and air quality aspects is needed for a better decay understanding.

Page 2 line 80 correct the capital letter line 89 correct That with This is done.

We changed the word.

Page 5 line 167 please change…this limiting can be overcome….with…this limit/aspect can be overcome.

We followed the recommendation, and instead of “limiting” it now says “limit”.

Page 6 line 192-193 please correct the sentence

The sentence was rewritten, from: “When reporting the results, the volume of artificial rain reported refers to the accumulated solution volume that has been irrigated to each specimen.

To “When reporting the results, “volume of artificial rain” refers to the total volume of solution irrigated on each specimen.” 

Page 6 line 195-200 again too long and heavy sentence. Make a full stop after ..samples collected ,monthly.

We appreciate the suggestion, and the paragraph was changed, from “However, in previous accelerated weathering publications, the rain has been prepared ei-ther with worse case scenarios [9] or the arithmetic average of samples collected monthly [52]; they both result in non-representative mass loss results, because the worst case scenario will overestimate the decay, whereas arithmetic average do not considerate the volume of collected rain: this means that, if in a month 5 mm of rain were collected, and in another one, 30 mm were sampled, they will both be considered the same “ to

However, in previous accelerated weathering publications, the rain has been prepared either with worse case scenarios [9] or the arithmetic average of samples collected monthly [52] but they both produce non-representative mass loss results. The worst-case scenario will overestimate the decay because conditions are regularly not that severe. On the other hand, the arithmetic average considerate all events the same regardless their volume: this means that, if 5 mm of rain were collected in a month, and in another one, 30 mm were sampled, both chemical compositions will be considered equally.”.

Page 7 line 220 please use if possible always the same concentration either N or M but do not mix it.

We thank the reviewer for highlighting the mistake. The concentration of H₂SO₄ was changed from 0.02 N to 0.01 M.

Page 8 line 261 – 262 something is not working….and confirm is (maybe you wanted to write IF?) mass, OP and….

We thank the reviewer for noticing the typo: it said “is” and should be “if”.

Page 8 line 266-267 please correct the sentence.  This test was done to ensure that there were..

The word “done” was included.

Page 8 line 284 check the size and format of table caption , be consistent . See also table 3 and 4 – correct the font and size

The font and size of Tables 2, 3 and 4 were corrected.

Page 10 Figure 4 authors need to have the same maximum unit for all the conditions. It is difficult to get the differences among them if you have different scales.

Figure 4 was changed: instead of presenting the results as three different plots, they are presented in the same one, and the scale is now consistent.

Page 11 line 328 check the caption of table 5

The caption was corrected, from “Mass, WAC, and OP before and after the accelerated weathering test.

To “Table 5. Mass, WAC, and OP averages and standard deviation before and after the accelerated weathering test.”

Page 11 line 332-333 have a look to the sentence, it is not clear . Line 333-337 again too long and unclear discussion. You need to separate the different comments and help the future reader to understand.

The paragraph was changed from:

“There are several reasons why OP and WAC are lower after the accelerated weathering: the pore network seems to be closing, probably mainly due to of the washing of clay minerals [60] that obstructs the pore network; this decrease cannot be because salt crystallization because the crystallized salts are soluble, and after the WAC test, salts were removed. It is important to note that this difference can be primarily because of the volume irrigated, although chemical composition plays an important role, because as it will be discussed in section Accelerated weathering, pH is indeed an important variable affecting decay; however, research is needed to distinguish if it is due to a higher ion concentration or because of pH, using for example, the salinity as an indirect measure of rain “cleanliness” in the future.

To:

“There are several reasons why OP and WAC diminish after the accelerated weathering. First, the volume irrigated to global group was higher than acid and no acid, and it can be the main cause of the difference for water produces all the mechanisms. Second, the pore network seems to be closing, probably mainly due to of the washing of clay minerals [60] that obstructs the pore network as it can be seen in figure 2 in which non-soluble particles were captured in the filters. The decrease in OP and WAC cannot be produced by salt crystallization because the crystallized salts are soluble, and after the WAC test, salts were removed. Finally, although precipitation volume is important, the chemical composition also plays an important role, because as it will be discussed in section Accelerated weathering, pH is indeed an important variable affecting decay; however, re-search is needed to distinguish if it is due to a higher ion concentration or because of pH, using for example, the salinity as an indirect measure of rain “cleanliness” in the future.” 

Page 14 Figure 6 Can you please explain me why here you have data for acid up to 5 on the X axis while in Figure 7 we do have all different scales on X axis? Honestly, I do not get how you can compare data or better behaviour among situation if you stop at different points

This is a core point of the discussion: to simulate one year precipitation, acid and no-acid groups should sum the volume of global group. To simulate a year, different volumes are needed for each group. That is the reason why all of them have different volumes: ten years of global precipitation are approximately 24 L, of no acid precipitation around 19 L and acid precipitation circa 5 L. Hence the need to develop damage functions depending on volume instead of time.

Reviewer 2 Report

Dear authors, the research work is very interesting from the perspective of accelerated aging experimental design applied to the different samples. However, it is confusing in its overall approach as the rest of the experimental methods used in the research are not adequately described, nor do they correspond properly to the presentation and expression of conclusions.

There is a significant error in Figure 4. It is worth noting that the graphs in Figure 4 cannot be properly compared due to the different scales presented by each of them (global difference, non-acid difference, or acid difference). They need to be redone and rescaled so that they can be visually compared. Furthermore, all three should be presented horizontally.

The graphs shown in figures 5, 6, and 7 must be represented mandatorily in an adimensional form, that is, in percentages. Otherwise, the variation of such a large sample cannot be understood.

Caption of Figure number 8, should be implemented with information relative to mean values and standard deviation incorporated in the plots using graphical symbols.

Additionally, conclusions should be numbered and indexed in order of descending relevance.

In the conclusions, it is stated that a significant mass loss has occurred due to washes of non-soluble compounds (mainly clay minerals), as highlighted in the text. This specific information must be scientifically justified with existing evidence or potential future experiments.

Similarly, it is indicated that cation exchange and salt crystallization should have occurred. This latter factor is a specific form of alteration in the stone materials, and therefore, graphical evidence of the supposed efflorescences must be presented.

Kind regards.

Author Response

We thank Reviewer for the feedback. It undoubtedly helps to improve and make the document more clear for future readers.

Dear authors, the research work is very interesting from the perspective of accelerated aging experimental design applied to the different samples. However, it is confusing in its overall approach as the rest of the experimental methods used in the research are not adequately described, nor do they correspond properly to the presentation and expression of conclusions.

We thank the reviewer for the comments. We shortened numerous sentences all over the document so now it is clearer what are the assumptions and how they can be conclusively demonstrated.

There is a significant error in Figure 4. It is worth noting that the graphs in Figure 4 cannot be properly compared due to the different scales presented by each of them (global difference, non-acid difference, or acid difference). They need to be redone and rescaled so that they can be visually compared. Furthermore, all three should be presented horizontally.

The plots were corrected and are now presented as a single one. The scale was also corrected.

The graphs shown in figures 5, 6, and 7 must be represented mandatorily in an adimensional form, that is, in percentages. Otherwise, the variation of such a large sample cannot be understood.

The figures 5 and 7 were corrected. Figure 6 was modified, to better illustrate why damage functions depending on a measurable parameter (as precipitation) is better than an arbitrary unit (as time).

Caption of Figure number 8, should be implemented with information relative to mean values and standard deviation incorporated in the plots using graphical symbols.

Figure 8 was modified so it now includes a legend regarding average, median and percentiles. Standard deviation was not included, because it can be calculated from the Inter Quartile Range (IQR).

Additionally, conclusions should be numbered and indexed in order of descending relevance.

We thank the reviewer for the suggestions. The conclusions are now numbered and hierarchized.

In the conclusions, it is stated that a significant mass loss has occurred due to washes of non-soluble compounds (mainly clay minerals), as highlighted in the text. This specific information must be scientifically justified with existing evidence or potential future experiments.

The text and Figure 2 were corrected. Now, there is a picture of the filters used, proving that some insoluble particles were washed away from the building material, although further XRD analysis is needed to confirm if they are clay minerals.

Similarly, it is indicated that cation exchange and salt crystallization should have occurred. This latter factor is a specific form of alteration in the stone materials, and therefore, graphical evidence of the supposed efflorescences must be presented.

Although a good suggestion, since all the specimens underwent immersion on deionized water, all possible efflorescences were dissolved, and no previous images were taken. However, the text was modified: now it only suggests them, and not conclude them as a decay mechanism, so in the future, analysis of the solution in which the specimens were submerged as well as XRD are needed to prove the presence and composition of any efflorescence.

Reviewer 3 Report

The paper entitled: Quantifying decay due to wet atmospheric deposition on basalt is a fascinating paper focused on the weather effect on basalt monuments.

The author used a specific method to evaluate the effect of humidity on this material and processed a huge amount of data.

However, in order to make this paper more sound for readers, I suggest reducing or combining the tables from annexes. There are too many and is difficult to follow them in the present version.

Also, I suggest adding some microscopic investigations (optical microscopy and/or SEM) for the basalt before and after the experiments. The presence or absence of some cracks, figures, or other damaging signs would be useful for this study.

Author Response

We appreciate greatly the comments and suggestions given by Reviewer.

The paper entitled: Quantifying decay due to wet atmospheric deposition on basalt is a fascinating paper focused on the weather effect on basalt monuments.

The author used a specific method to evaluate the effect of humidity on this material and processed a huge amount of data.

However, in order to make this paper more sound for readers, I suggest reducing or combining the tables from annexes. There are too many and is difficult to follow them in the present version.

We thank the reviewer for the suggestion. Information of two annexes was already included in the text (although less detailed), so to avoid confusion, they were removed.

Also, I suggest adding some microscopic investigations (optical microscopy and/or SEM) for the basalt before and after the experiments. The presence or absence of some cracks, figures, or other damaging signs would be useful for this study.

Although the suggestions are a great way to strengthen the focus of the paper, unfortunately they were not performed before the test, and the macrophotographs do not present conclusive signs of decay, as evident cracks, or higher porosity. However, it was included in Figure 2 a photograph of the filters to better demonstrate the washing of insoluble compounds, probably clay minerals, that need to be further analyzed by XRD means.

Reviewer 4 Report

GENERAL COMMENTS

The study focuses on the quantification of damage due to wet atmospheric deposition on a red basalt used in several important heritage constructions in Mexico. 

The subject of the work is interesting, but there are a few issues with the experimental details and discussion, as specified below.

The technical English requires revision.

SPECIFIC COMMENTS

Line 109: I believe the authors mean “stone ashlar” instead of “stone brick,” please, correct.

Fig. 2 – The images in this figure show that the porosity is highly heterogeneous among specimens. This aspect seems to not have been taken into account for defining the size of the specimens. This issue can cause biased results.

Eq. (1) Please, state the meaning of “L”

Line 185: What do the authors mean with “natural drying”? Please, state the conditions (temperature, relative humidity).

Line 197: When the authors write “to try to monitor decay” – do you mean by measuring the mass loss? This would be possible but only through collection of stone debris from the specimens.

Lines 219-220: I cannot understand why “the pH was corrected.” Can you please explain better?

Given that the Accelerated weathering test is complex and not common, I suggest presenting a scheme summarizing the most important steps.

Line 230 – I think that this subchapter should be named differently, e.g. Analysis of the artificial rain used in the accelerated weathering test 

I think that the use of the Water absorption capacity and the Open porosity as properties to monitor decay must be carefully considered because they are influenced by the mass loss and by deposition of recrystallised products that may have low solubility. The results presented (Fig. 4) illustrate this issue.

Table 5: I suggest presenting the average and standard deviation, otherwise the table is a visual burden to the reader.

Lines 353-354: I do not agree with this explanation, I think that the large mass difference is rather associated with the porosity heterogeneity, as I mentioned previously. In the future, I suggest using bigger samples to counteract this issue.

Table 8: I do not understand the usefulness of this table. 

The technical English requires revision.

Author Response

We greatly appreciate the suggestions to the Reviewer, for it helps to improve the research.

The study focuses on the quantification of damage due to wet atmospheric deposition on a red basalt used in several important heritage constructions in Mexico. 

The subject of the work is interesting, but there are a few issues with the experimental details and discussion, as specified below.

We thank the Reviewer for the feedback. It helps the document to improve in general.

Line 109: I believe the authors mean “stone ashlar” instead of “stone brick,” please, correct.

 We agree with the Reviewer and have changed “stone brick” for “stone ashlar”.

Fig. 2 – The images in this figure show that the porosity is highly heterogeneous among specimens. This aspect seems to not have been taken into account for defining the size of the specimens. This issue can cause biased results.

Porosity is indeed highly heterogeneous, which is unavoidable due to materials’ genesis. Since it was not possible to calculate the dimensions needed for each specimen to have a similar mass, we decided to use apparent volume as the main variable to prepare the specimens for accelerated weathering. This was acknowledged on lines 113-116.

Eq. (1) Please, state the meaning of “L”

“L” stands for Liters. However, to avoid confusion, the units of the Equation are now inside brackets.

Line 185: What do the authors mean with “natural drying”? Please, state the conditions (temperature, relative humidity).

For natural drying we mean the conditions of the laboratory, which ranged from 16°C to 23°C and relative humidity from 40% to 60%. They were not controlled to better simulate the outdoor exposure in which temperature and relative humidity changes are severer. However, it is a good suggestion for future studies to control both variables. This was all addressed on lines 179-188.

Line 197: When the authors write “to try to monitor decay” – do you mean by measuring the mass loss? This would be possible but only through collection of stone debris from the specimens.

We exactly mean periodically measuring the mass, which would be very difficult for the whole building. That is the reason why we note the need for another variable to monitor decay and propose an invasive and non-destructive methodology through Karsten tube. Using debris is a good idea, but still, they would not allow the monitoring of mass through time of the building because other mechanisms like lixiviation also produces mass loss without waste.

Lines 219-220: I cannot understand why “the pH was corrected.” Can you please explain better?

Of course. The objective chemical compositions were calculated from measurements, so they could not be replicated exactly because some ions would interact with each other, producing a different pH than expected. An excellent example is acid group, whose concentration of NH₄⁺ and Ca²⁺ can neutralize acidity (Ramírez-Lara et al, 2010; Sosa et al, 2023) and hence the need to correct pH.

Given that the Accelerated weathering test is complex and not common, I suggest presenting a scheme summarizing the most important steps.

We apologize with the reviewer because there was a mistake in the document: it said “step” and it should have said “cycle”. Hence, all over the document, the words were changed (i.e. lines 179, 182, 190).

A Scheme (Figure 2) was included to clarify the experimental design. Besides, lines 179-188 explain details of conditions for each simulation step. The irrigation time was different for each group because although it would undoubtedly affect the wetting time of the material, changing the artificial rain solution flow would have caused another variable: the mechanical effect that the water drop have on the material. To avoid it, the irrigation time to the same flow was proposed.

Line 230 – I think that this subchapter should be named differently, e.g. Analysis of the artificial rain used in the accelerated weathering test 

We thank the Reviewer for the suggestion. However, the analytical method used was Ion Chromatography, and it was performed both on collected rain samples and the samples collected from accelerated weathering tests, so this section was not renamed. Still, in the discussion, the section name was changed from “3.5 Ion Chromatography” to “3.5. Ion Chromatography used in analysis of accelerated weathering test samples”.

I think that the use of the Water absorption capacity and the Open porosity as properties to monitor decay must be carefully considered because they are influenced by the mass loss and by deposition of recrystallised products that may have low solubility. The results presented (Fig. 4) illustrate this issue.

We acknowledge the limits that open porosity and water absorption capacity have to be used as a decay variables. As the reviewer states, it is true that they are influenced by several mechanisms including recrystallization of salts, expansion/contraction of clay minerals, cation exchange, or lixiviation to name a few, all of them resulting in mass loss. However, and since one of the objectives of the research is to find a different variable than mass change to monitor decay, they were proposed, but we do not propose to use it instead of mass loss yet. Our intention is to contribute to the development of a different variable that can be measured in situ on built heritage when no mass can be recorded.

Table 5: I suggest presenting the average and standard deviation, otherwise the table is a visual burden to the reader.

 We agree with the comment, and substitute table 5 for a table in which only average and standard deviation are presented.

Lines 353-354: I do not agree with this explanation, I think that the large mass difference is rather associated with the porosity heterogeneity, as I mentioned previously. In the future, I suggest using bigger samples to counteract this issue.

We appreciate the comment. However, and although the large mass difference is associated with porosity heterogeneity, what was done was a statistical comparison standardizing the mass of each sample before and after the accelerated weathering test. Since it is standardized, and each specimen is compared with itself through a t-paired test, so the difference, although affected by porosity, can be considered statistically significant. Still, it is a great advice to use bigger samples in future studies.

Table 8: I do not understand the usefulness of this table. 

This table is useful to have a glance on possible significant differences in overall chemical composition. Since the number of variables was big (one per ion), we analyzed the whole chemical composition as a vector and compare among them. When the Alternative hypothesis was found, that means that the chemical composition of the lixiviate sample was significantly different than the artificial rain irrigated.

Bibliography

  1. Borrelli, Porosity. ARC Laboratory Handbook, Roma: ICCROM, 1999.

Ramírez Lara, E.; Miranda Guardiola, R.; Gracia Vásquez, Y.; Balderas Rentería, I.; Bravo Álvarez, H.; Sosa Echeverría, R.; Sánchez Álvarez, P.; Alarcón Jiménez, A.; Torres M.C. and Kahl, J. (2010). Chemical composition of rainwater in northeastern México. Atmósfera, 23(3), 213-224. Recuperado en 26 de julio de 2023, de http://www.scielo.org.mx/scielo.php?script=sci_arttext&pid=S0187-62362010000300001&lng=es&tlng=en.

Rodolfo Sosa Echeverría, Ana Luisa Alarcón Jiménez, María del Carmen Torres Barrera, Pablo Sánchez Alvarez, Elías Granados Hernandez, Elizabeth Vega, Mónica Jaimes Palomera, Armando Retama, David A. Gay (2023) Nitrogen and sulfur compounds in ambient air and in wet atmospheric deposition at Mexico city metropolitan area, Atmospheric Environment, Volume 292, 2023, 119411, ISSN 1352-2310, https://doi.org/10.1016/j.atmosenv.2022.119411.

Round 2

Reviewer 2 Report

Dear authors the corrections were done properly.

Author Response

We appreciate the final opinion of the reviewer. Undoubtedly, all observations during the first stage of the review served to improve and enrich the article.

Reviewer 4 Report

The authors have conducted a thorough revision of the manuscript, which has improved significantly its quality. 

Regarding my comments on the dimensions of the samples used, I thank the response from the authors, however, I think that this aspect should be mentioned in the manuscript, at least as a consideration in the Conclusions section for future work. Indeed, the dimensions should account for the heterogeneity in such a way that the samples have a similar mass.

The technical English can still be improved, for instance: 

Line 39: “The influence of atmosphere on the decay of heritage buildings decay (…)” reads poorly. Suggestion:  “The influence of atmospheric agents on the decay of heritage buildings (…)”

The technical English can still be improved, for instance: 

Line 39: “The influence of atmosphere on the decay of heritage buildings decay (…)” reads poorly. Suggestion:  “The influence of atmospheric agents on the decay of heritage buildings (…)”

Author Response

Responses to Reviewer 4

We thank the Reviewer for the comments that helped to improve the manuscript quality.

Regarding my comments on the dimensions of the samples used, I thank the response from the authors, however, I think that this aspect should be mentioned in the manuscript, at least as a consideration in the Conclusions section for future work. Indeed, the dimensions should account for the heterogeneity in such a way that the samples have a similar mass.

We thank the reviewer for enrichening the methodology. We added lines 369-371

“However, in the future, when a materials’ porosity is heterogeneous, different specimen dimensions may be considered.” 

 Conclusion I was modified. It stated (line 504):
“The methodology employed is suitable for monitoring decay”

Now it reads:

“The methodology employed is suitable for monitoring decay, although changing specimen dimensions can be changed to diminish the effect of porosity heterogeneity.”

Line 39: “The influence of atmosphere on the decay of heritage buildings decay (…)” reads poorly. Suggestion:  “The influence of atmospheric agents on the decay of heritage buildings (…)”

We thank the reviewer and have changed line 39. Now it is:

“The influence of atmospheric agents on the decay of heritage buildings is a topic that has been studied for over 60 years.”
